# Position: Stop Treating 'AGI' as the North-star Goal of AI Research

**Borhane Blili-Hamelin** [* 1]   **Christopher Graziul** [* 2]   **Leif Hancox-Li** [† 3]   **Hananel Hazan** [† 4]
**El-Mahdi El-Mhamdi** [‡ 5 6]   **Avijit Ghosh** [‡ 7 8]   **Katherine Heller** [‡ 9]   **Jacob Metcalf** [‡ 10]   **Fabricio Murai** [‡ 11]
**Eryk Salvaggio** [‡ 12]   **Andrew Smart** [‡ 9]   **Todd Snider** [‡ 13]   **Mariame Tighanimine** [‡ 14]   **Talia Ringer** [§ 15]
**Margaret Mitchell** [§ 7]   **Shiri Dori-Hacohen** [§ 8]

## Abstract

The AI research community plays a vital role in shaping the scientific, engineering, and societal goals of AI research. In this position paper, we argue that focusing on the highly contested topic of 'artificial general intelligence' ('AGI') undermines our ability to choose effective goals. We identify six key traps—obstacles to productive goal setting—that are aggravated by AGI discourse: Illusion of Consensus, Supercharging Bad Science, Presuming Value-Neutrality, Goal Lottery, Generality Debt, and Normalized Exclusion. To avoid these traps, we argue that the AI research community needs to (1) prioritize **specificity** in scientific, engineering, and societal goals, (2) center **pluralism** about multiple worthwhile approaches to multiple valuable goals, and (3) foster innovation through greater **inclusion** of disciplines and communities. Therefore, the AI research community needs to **stop treating "AGI" as the north-star goal of AI research**.

## 1. Introduction

How can we ensure that AI research goals serve scientific, engineering, and societal needs? What constitutes good science in AI research? Who gets to shape AI research goals? What makes a research goal legitimate or worthwhile? In this position paper, we argue that a widespread emphasis on AGI threatens to undermine the ability of researchers to provide well-motivated answers to these questions.

Recent advances in large language models (LLMs) have sparked interest in "achieving human-level 'intelligence'" as a "north-star goal" of the AI field (McCarthy et al., 1955; Morris et al., 2024). This goal is often referred to as "artificial general intelligence" ("AGI") (Chollet, 2024a; Tibebu, 2025). Yet rather than helping the field converge around shared goals, AGI discourse has mired it in controversies. Researchers diverge on what AGI is and assumptions about goals and risks (Summerfield, 2023; Morris et al., 2024; Blili-Hamelin et al., 2024). Researchers further contest the motivations, incentives, values, and scientific standing of claims about AGI (Gebru & Torres, 2024; Mitchell, 2024; Ahmed et al., 2024; Altmeyer et al., 2024). Finally, the building blocks of AGI as a concept—intelligence and generality—are contested in their own right (Gould, 1981; Anderson, 2002; Hernández-Orallo & Seán Ó hÉigeartaigh, 2018; Cave, 2020; Raji et al., 2021; Alexandrova & Fabian, 2022; Blili-Hamelin & Hancox-Li, 2023; Hao, 2023; Guest & Martin, 2024; Paolo et al., 2024; Mueller, 2024).

Building on prior work on the ambiguity between exploratory and confirmatory research in ML (Herrmann et al., 2024), unscientific performance claims (Altmeyer et al., 2024), SOTA-chasing (Raji et al., 2021; Church & Kordoni, 2022), homogenization of research approaches (Kleinberg & Raghavan, 2021; Fishman & Hancox-Li, 2022; Bommasani et al., 2022), the values embedded in ML research (Birhane et al., 2022b), and more, our account identifies key obstacles to productive goal setting in AI research—**traps**.[1] We provide them here as a diagnosis of problems in goal-setting

---

[*]Equal contribution & Lead contributors [†]Top contributors [‡]Contributors (alphabetical order) [§]Senior contributors [1]AI Risk and Vulnerability Alliance, NY, USA [2]University of Chicago, IL, USA [3]Vijil, CA, USA [4]Tufts University, MA, USA [5]Calicarpa, Switzerland [6]Ecole Polytechnique, France [7]Hugging Face, NY, USA [8]University of Connecticut, CT, USA [9]Google, CA, USA [10]Data & Society Research Institute, NY, USA [11]Worcester Polytechnic Institute, MA, USA [12]Rochester Institute of Technology, NY, USA [13]Eberhard Karls Universität Tübingen, Tübingen, Germany [14]Conservatoire national des arts et métiers, Lise-CNRS, France [15]University of Illinois Urbana-Champaign, IL, USA. Correspondence to: Borhane Blili-Hamelin <borhane@avidml.org>, Christopher Graziul <graziul@uchicago.edu>, Talia Ringer <tringer@illinois.edu>, Margaret Mitchell <meg@huggingface.co>, Shiri Dori-Hacohen <shiridh@uconn.edu>.

*Proceedings of the 42nd International Conference on Machine Learning*, Vancouver, Canada. PMLR 267, 2025. Copyright 2025 by the author(s).

---

[1]Our terminology parallels Selbst et al. (2019) on fairness.

that we believe are normatively worth addressing, but that the AGI narrative makes difficult to overcome. To avoid these traps, **we posit that communities should stop treating AGI as the north-star goal[2] of AI research**.

An overarching theme in our discussion is the research community's *unique responsibility to help distinguish hype from reality*. The outputs of AI research are deployed as real-world products at a staggering pace, in proliferating contexts, affecting billions of people. This warrants urgent work on trusted, evidence-based answers to questions about the scientific, engineering, and societal merits of AI tools. As argued by the U.N.'s AI Advisory Body, there is "an overwhelming amount of information... making it difficult to decipher hype from reality. This can fuel confusion, forestall common understanding and advantage major AI companies at the expense of policymakers, civil society and the public" (United Nations, 2024). Our position paper addresses this theme: Each of the six traps in our account is an obstacle to distinguishing hype from reality.

A secondary theme in our discussion is the relationship between people and technology. Ultimately, we argue that instead of a single north-star goal, the AI community needs to pursue *multiple specific* scientific, engineering, and societal goals. If building consensus around an alternative unifying goal proves useful, we propose the goal of *supporting and benefiting human beings*.

In the next section, we examine six 'traps' in AI research—obstacles to productive goal setting (§2). We argue that AGI discourse reinforces and amplifies each problem. Subsequently, we provide three recommendations for avoiding these traps (§3): specificity of goals; pluralism of goals and approaches; and more inclusive goal setting. We conclude by offering a rebuttal against an *alternative view* (§4): that AGI should remain the north-star goal of the field.

Because the contested nature of AGI is a central theme in the present paper, we avoid providing our own definition for the term. Instead, we provide example definitions throughout the present discussion, as well as a table of illustrative definitions in Appendix A.[3] We detail how AGI currently serves as a north-star goal in Appendix B.

---

[2]Sailors who navigate by the astronomical North Star use it to orient their travels toward a desired destination on Earth. With AGI, some researchers are using AGI as a "guiding star" to orient their AI research "travels" towards. Other researchers, however, are actually hoping and working towards the goal of "arriving" at AGI. Our paper argues against both of these approaches.

[3]Arguably, many of the concerns we raise about AGI apply to other terms used to refer to future forms of AI, such as "powerful AI" (Amodei, 2024) and "transformative AI" (Gruetzemacher & Whittlestone, 2022). Ultimately, as we revisit in §4, our account can be viewed as critically interrogating north-star goals more generally.

## 2. Traps

We examine six key *traps* that hinder the research community's ability to set worthwhile goals. We argue that each is aggravated by AGI narratives.

The problems we discuss are highly interrelated. For instance, SOTA-chasing, discussed in relationship to the role of misaligned incentives in shaping goal setting (§2.4), also has implications for bad science (§2.2). Similarly, the problem of the lack of consensus about AGI (§2.1) is a theme that recurs throughout. We do not intend the traps to be mutually exclusive. Rather, our goal for each trap is to provide distinct and useful insights for mitigating failure modes in productive goal setting.

### 2.1. Illusion of Consensus

*Using shared term(s) in a way that gives a false impression of consensus about goals, despite goals being contested*

The popular use of the term "AGI" (Grossman, 2023; IBM, 2023; Holland, 2025) creates a sense of familiarity, giving the illusion that there is a shared understanding on what AGI is, and broad agreement on research goals in AGI development. However, there are vastly different opinions on what the term AGI refers to, what an AGI research agenda looks like, and what the goals in AGI development are. Left unchecked, this illusion obstructs explicit engagement on what the goals of AI research are and should be.

From popular discourse to research papers to corporate marketing materials, the vast majority of references to AGI fall into this trap when they uncritically cite claims about so-called AGI. For examples of uncritical media claims, see Grossman (2023) and IBM (2023); see Altmeyer et al. (2024) for examples of overhyped research.[4] Summerfield (2023) summarizes the issue: "AI researchers hope to discover how to build AGI. The problem is that nobody really knows exactly what an AGI would look like." Mueller (2024) calls AGI "a meaningless concept, an emperor with no clothes." Blili-Hamelin et al. (2024) identify multiple types of disagreement among definitions of AGI or human-level AI. The contested nature of AGI as a goal is even more acute in critiques of AGI concepts (e.g., Altmeyer et al., 2024; Mueller, 2024; Van Rooij et al., 2024).

Beyond AGI, AI research is rife with topics that involve disagreement about goals, values, and concepts. For example, Mulligan et al. (2016) argue that *privacy* should be understood as an "essentially contested concept." They argue lack of agreement about the meaning and significance of privacy is not merely a matter of confusion—rather, dis-

---

[4]Some researchers who advocate for AGI as a goal have avoided the Illusion of Consensus trap (§2.1); e.g., Morris et al. (2024) explicitly call for investigating disagreements about goals, predictions, and risks that underpin prominent accounts of AGI.

agreement and contestation are desirable features that enable privacy to adapt to changing technical and social contexts. Similarly, there is now widespread acceptance *fairness* should be understood as a contested topic, not only admitting incompatible mathematical formalizations but also incompatible values, worldviews, and theoretical assumptions (Friedler et al., 2021; Jacobs & Wallach, 2021).

In suggesting this trap, we do not presume that contested topics are inherently problematic. Rather, we argue that when dealing with the important question of the goals of AI research, the significant disagreements that surround AGI should be embraced as signals of conflicting values.

### 2.2. Supercharging Bad Science

*Worsening current problems with bad science in AI due to poorly defined concepts and experimental procedures*

Research that produces reliable empirical knowledge about AI is vital to public interest decisions about AI's potential for societal and environmental benefit and harm. Yet, many experts have noted a pervasive lack of scientific grounding in AI research (Hullman et al., 2022; Raji et al., 2022; Sloane et al., 2022; Suchman, 2023; Guest & Martin, 2024; Narayanan & Kapoor, 2024; United Nations, 2024; Van Rooij et al., 2024; Widder & Hicks, 2024). We argue that vagueness in AGI discourse exacerbates existing problems with the scientific validity of AI research.

**Problem 1: Underspecification and external validity.** One problem with the pursuit of AGI as a concrete goal is **underspecification** (D'Amour et al., 2022), where *lack of specificity in goals or concepts leads to cascading epistemic problems*, including irrefutability, lack of external validity, flawed experimental design, and flawed evaluation. These common problems in AI research are worsened in the AGI context by the lack of scientifically grounded definitions of AGI (§2.1).

Underspecification of learning goals also undermines *external validity*—the question of whether a measurement corresponds to the real-world phenomenon it's supposed to capture. A good example is the debate about whether "language understanding" benchmarks actually measure language understanding (Jacobs & Wallach, 2021; Liao et al., 2021).

External validity is also relevant when researchers equate human faculties with model proxies (Hullman et al., 2022), such as claiming that a model "capable of linking specific objects with more general visual context" is evidence of "imagination" (Fei et al., 2022). This rhetorical move is enabled by using colloquial terms like "imagination" without considering whether it corresponds to the human faculty. Altmeyer et al. (2024) likewise critiques Gurnee & Tegmark (2024) for inflated claims enabled by

the vagueness of the term "world model". Underspecified goals trickle down into many areas of experimental design, such as learning pipelines, evaluation metrics, tasks, representations, and methods.

External validity is also undermined when researchers claim to measure concepts from other fields, like intelligence. The fields of psychology, neuroscience, and cognitive science have studied human intelligence for generations, yet even they lack consensus on what "intelligence" is (Gopnik, 2019; Hao, 2023). Conversely, AI research is no longer concerned with modeling human cognition (Guest & Martin, 2024; Van Rooij et al., 2024). Instead, AI developers define "intelligence" on their own terms, privileging definitions convenient for benchmarking or selling products (§2.4), while benefiting from historically positive connotations of the term "intelligence".

**Problem 2: Ambiguity between science and engineering.** Another problem with the pursuit of AGI is *confusion between science and engineering* (Agre, 2014; Hutchinson et al., 2022; Altmeyer et al., 2024). As Hullman et al. (2022) point out, a "typical supervised ML paper" (e.g., one that reports accuracy metrics on a benchmark) is often just an "engineering artifact", a tool attached to performance claims that cannot be refuted because of replication challenges. Altmeyer et al. (2024) argue that this ambiguity between science and engineering means rigorous hypothesis testing with "specific conditions and considering effect sizes" is often omitted, with results often presented as "engineering achievements" without specifying *precisely* what is being tested, relevant hypotheses, and what effect sizes would constitute substantial findings.

This ambiguity invites experimenter and confirmation biases, since researchers are incentivized to "pay little or no attention to competing hypotheses or explanations" or "[fail] to articulate a sufficiently strong null hypothesis," (Altmeyer et al., 2024). Confusion between science and engineering also manifests when it is unclear if a study is pursuing scientific goals—of explanation, hypothesis confirmation, etc.—or goals of specific engineering applications—e.g., a proof-of-concept (Hutchinson et al., 2022). This exacerbates questions about external validity: without clear and specific experimental goals, it is easier to provide post-hoc interpretations of experiments that "support" a wide variety of goals (§2.4).

**Problem 3: Ambiguity between confirmatory and exploratory research.** The ambiguity between engineering and scientific methodology is related to another problem: *confusion between confirmatory and exploratory research* (Bouthillier et al., 2019; Herrmann et al., 2024). Herrmann et al. (2024) state that confirmatory research "aims to test preexisting hypotheses to confirm or refute existing theories [while] exploratory research is an open-

ended approach that aims to gain insight and understanding in a new or unexplored area." They go on to argue that "most current empirical machine learning research is fashioned as confirmatory research while it should rather be considered exploratory" and that experiments are "set up to *confirm* the (implicit) hypothesis that the proposed method constitutes an improvement" (emphasis theirs). By implicitly conflating exploratory analysis with confirmatory research, "exploratory findings have a slippery way of 'transforming' into planned findings as the research process progresses" (Calin-Jageman & Cumming, 2019). Using the vague and contested concept of AGI to frame confirmatory claims worsens this problem, as it makes it harder to figure out *what* is being claimed.

## 2.3. Presuming Value-Neutrality

*Framing goals as purely technical or scientific, when they are in fact laden with political, social, or ethical values*

Presuming Value-Neutrality occurs when technical or scientific goals become disconnected from their **value-laden** assumptions: aspects of AI research that are—and should be—informed by political, social, and ethical considerations. The AI research community has recently begun examining these value-laden assumptions (Shilton, 2018; Broussard et al., 2019; Abebe et al., 2020; Blodgett et al., 2020; Costanza-Chock, 2020; Denton et al., 2020; 2021; Dotan & Milli, 2020; Birhane & Guest, 2021; Green, 2021; Scheuerman et al., 2021; Viljoen, 2021; Birhane et al., 2022b; Bommasani, 2023; Fishman & Hancox-Li, 2022; Hutchinson et al., 2022; Mathur et al., 2022; Blili-Hamelin & Hancox-Li, 2023; Blili-Hamelin et al., 2024; Zhao et al., 2024).

When efforts to define AGI and related concepts do not explicitly examine the societal goals and values embedded in their definitions, they fall into the Presuming Value-Neutrality trap. Examples include proposals for "universal intelligence" (Legg & Hutter, 2007; Hernández-Orallo et al., 2014).

The pursuit of value-neutral approaches echoes debates about psychometric views of human intelligence. Intelligence, like "health," and "well-being," inherently carries normative assumptions about which behaviors or abilities are desirable (Anderson, 2002; Alexandrova & Fabian, 2022). Researchers fall into the Presuming Value-Neutrality trap by sidestepping these value-laden dimensions (Anderson, 2002; Cave, 2020; Blili-Hamelin & Hancox-Li, 2023). Warne & Burningham (2019) exemplify this by advocating for purely statistical definitions of intelligence, precisely because cultural definitions vary.

Value-laden assumptions within concepts like AGI drive legitimate disagreement about their meaning, reflecting di-

vergent societal goals (Blili-Hamelin et al., 2024). This makes consensus on AGI challenging, as it requires alignment on political, social, and ethical priorities. Similar disagreements affect related concepts like AI (Cave, 2020; Blili-Hamelin & Hancox-Li, 2023), "human-level AI", "superintelligence", and "strong AI", reinforcing the Illusion of Consensus trap (§2.1).

## 2.4. Goal Lottery

*Adopting goals which are not adequately justified by scientific, engineering, or social merit, but instead on the basis of incentives, circumstances, or luck*

Researchers have studied the role of socioeconomic factors, trends, and circumstantial factors in shaping AI research. For instance, Hooker (2021) has argued that a form of hardware lottery—the greater availability of hardware with strengths in parallel processing—was key to the resurgence of deep learning in the 2010s.[5] Similarly, researchers have examined the role of incentives, socioeconomic factors, and hype cycles in AI research (Raji et al., 2022; Delgado et al., 2023; Sartori & Bocca, 2023; Widder & Nafus, 2023; Gebru & Torres, 2024; Hicks et al., 2024; Narayanan & Kapoor, 2024; Wang et al., 2024). With this trap, we focus on cases where lotteries (luck) or incentives drive the adoption of unjustified goals—goals that are inadequately supported by scientific, engineering, or societal merit.

Consider AGI definitions centered on economic value, like OpenAI's emphasis on "outperform[ing] humans at most economically valuable work" (OpenAI, 2018). The primacy of economic value for setting AI research goals is contentious from both engineering and societal perspectives. Such definitions create misalignment between incentives and justifications by reducing complex societal, engineering, and scientific considerations to purely economic metrics.

Another example is benchmark SOTA-chasing—pursuing top scores on popular benchmarks (Bender et al., 2021; Raji et al., 2021; Church & Kordoni, 2022; Hullman et al., 2022). Despite strong professional incentives encouraging this practice, it lacks scientific, engineering, and societal justification. Benchmarks poorly reflect model performance in real application contexts because of problems like data leakage, overfitting to benchmarks, and data heterogeneity (El-Mhamdi et al., 2021; 2023; Hanneke & Kpotufe, 2022; Balloccu et al., 2024; Xu et al., 2024; Zhang et al., 2024). In short, the measurement method lacks *external validity*. Yet the practice persists

---

[5]On similar lottery or path dependence effects, see Liebowitz & Margolis (1995); Peacock (2009); Dehghani et al. (2021); Fishman & Hancox-Li (2022); Rossbach (2023); Bauer & Gill (2024); Hooker (2024).

due to reputational and financial rewards, demonstrating misalignment between incentivized goals and their actual merits.

The dynamics of goal lotteries are also visible in the story of the multi-decade neglect of deep learning architectures. In this case, a research agenda was sidelined for reasons that eventually proved to be misguided from an engineering, scientific, or societal perspective (e.g., due to "gatekeeping" effects against less popular research agendas; see Siler et al., 2015). Meanwhile, the AI industry went all in on the expert systems "bubble" (Haigh, 2024). *Reductions in diversity within* contemporary AI research can be a sign that similar mistakes are at play (§2.6). Some recent initiatives to counter homogenization (Chollet et al., 2024) rely on operationalizing AGI through benchmarks.[6] In practice, they end up as yet another benchmark: incentivizing SOTA-chasing, supercharged by intense media and marketing attention (Jones, 2025). For this reason, we remain somewhat skeptical of whether approaches like ARC (Chollet, 2019) outweigh the negative consequences of news-cycle-accelerated SOTA-chasing.

## 2.5. Generality Debt

*Relying on the generality or flexibility of tools to postpone crucial engineering, scientific, or societal decisions*

AGI definitions differ on how much "generality" is desirable (Blili-Hamelin et al., 2024). This indicates a lack of clarity and consensus about the goals of AI research, forming a trap that (a) encourages suboptimal science/engineering practices (related to points made in 2.2); (b) suppresses important social/ethical questions about which research directions are worth pursuing. We term this trap "Generality Debt" to parallel technical debt (Sculley et al., 2014): it delays the work that needs to be done as part of AI research which, if left undone, takes more work to address in the future.

This trap includes the appeal to many different notions of generality at play in machine learning: (1) variety of tasks (Hernández-Orallo & Seán Ó hÉigeartaigh, 2018); (2) capability to be trained for "any task" vs. ability to perform many predefined tasks (Hernández-Orallo & Seán Ó hÉigeartaigh, 2018); (3) whether the task or data distribution the model is being evaluated on is "seen" or "unseen" (i.e., available, or not, to the model during its training phase) (Altmeyer et al., 2024); (4) variety of data in model input/output, such as structured vs unstructured, modality, etc.; (5) whether the performance of the model reflects "performance considered 'surprising' to humans" (Altmeyer et al., 2024); (6)

variety of goals; (7) ability to "accept a general language for the problem statement" (Newell & Ernst, 1965); and (8) having a "general" internal representation (Newell & Ernst, 1965; McCarthy & Hayes, 1981).

As Paolo et al. (2024) note, despite the multiple possible meanings of "generality", most papers do not define generality even if it is central to their argument. Without formal definition, assessing or improving generalization becomes challenging. Assuming that "generalization" is desirable while acknowledging its poor definition is misguided. We should first define specific, measurable properties before arguing that they are desirable.

**Without proper definition, the value of generality remains unclear.** Different types of generality support different future visions, raising unexplored questions about their relative importance. Further, vague definitions of generality lead to bad science and engineering (§2.2). For example, Altmeyer et al. (2024) note how the pursuit of generality has led to vague task specifications. In parallel, Gebru & Torres (2024) argue that some conceptions of AGI contravene good engineering practices: it is hard to test the functionality of systems under "standard operating conditions" if the system is advertised as a "universal algorithm for learning and acting in any environment."

Aiming to achieve certain forms of generality could also mean making a tradeoff with ecological validity, as argued by Saxon et al. (2024). They argue that, in practice, "holistic" benchmarks tend to be a collection of disparate specific benchmark tasks, meaning that they have task-level construct validity. However, these tasks do not always match with *user-relevant capabilities*. Methodological challenges to achieving such capabilities may or may not be overcome, in time, given innovative solutions, but the pursuit of AGI assumes such challenges are surmountable. Methodological issues often inspire novel solutions, but we cannot assume a solution will be found, nor can those pursuing AGI. Further, strategies for pursuing AGI may introduce or reveal new challenges to achieving user-relevant capabilities.

Finally, the vagueness around "generality" is also an ethical concern, as it sidesteps normative questions about *which types of generality merit pursuit* and obscures implicit decisions about how to prioritize different research directions.

## 2.6. Normalized Exclusion

*Excluding communities and experts from shaping goals*

The negative consequences of exclusion in AI have been extensively discussed, both in terms of how it affects product quality and model performance (e.g., Obermeyer et al., 2019), and also how it harms people (e.g., Buolamwini & Gebru, 2018; Shelby et al., 2023; Whitney & Norman, 2024). We argue that AGI discourse

---

[6]Chollet proposes that "We will have AGI when creating [benchmarks 'that are easy for humans, yet impossible for AI'] becomes outright impossible" (2024b).

aggravates problems of exclusion.

**Problem 1: Excluding communities.** Many communities are left out of meaningful participation in shaping the goals of AI research (Delgado et al., 2023). Excluding communities causes serious harm, especially to minoritized communities (Pierre et al., 2021); it also undermines the utility of end products, reduces model performance, oversimplifies technical challenges (Kierans et al., 2025), and can impede innovation (e.g., Burt, 2004). For instance, the infamous case of facial recognition engines—such as those used by Google, Apple, and Meta—mistaking Black people for gorillas (BBC News, 2015) is still occurring more than 8 years after the problem was first identified (Appelman, 2023; Grant & Hill, 2023), with downstream impacts on surveillance and law enforcement (Jones, 2020; Pour, 2023). Similarly, selective forms of inclusion in data annotation raise ethical and practical concerns about downstream effects (Wang et al., 2022; Bertelsen et al., 2024). Other examples of exclusion or inclusion of communities impacting performance are plentiful (see Buolamwini & Gebru, 2018; Young et al., 2019; Raji et al., 2020; Andrews et al., 2024; Bergman et al., 2024; Salavati et al., 2024; Weidinger et al., 2024). Excluding communities from meaningful feedback also undermines societal goals, such as fostering collective legitimacy through accountability to impacted communities (Schulz et al., 2002; Mikesell et al., 2013; Costanza-Chock, 2020; Birhane et al., 2022a; Young et al., 2024).

The recent prominence of AGI discourse intensifies the existing problem of community exclusion in AI research (Frank et al., 2017; Kelly, 2024). Many proponents of AGI envision a future where AI systems perform an extraordinary range of tasks for countless communities. However, research and design processes fall short of the inclusiveness demanded by this ambitious vision. For example, December 2024 reporting suggests that OpenAI and Microsoft "signed an agreement last year stating OpenAI has only achieved AGI when it develops AI systems that can generate at least $100 billion in profits" (OpenAI, 2025d; Zeff, 2025), a stark departure from OpenAI's public definition of AGI (OpenAI, 2018). As several authors argue, economic value is not the only type of desirable value (Agrawal et al., 2022; Dulka, 2022; Harrigian et al., 2023; Morris et al., 2024; Pierson et al., 2025). The economic definition helps guide the engineering decisions of OpenAI. However, it is questionable whether an emphasis on profits will lead to the most beneficial or useful end products or to meaningful consensus about goals, especially for minoritized groups.

**Problem 2: Excluding disciplines.** From application domains (e.g., medicine (Obermeyer et al., 2019), finance (Cao, 2022), cybersecurity (Salem et al., 2024), learning (Leong & Linzen, 2024)) to the practices involved in building AI—data annotation, qualitative and quantitative methods, domain expertise, computer science, and many more (Wang et al., 2022; Bertelsen et al., 2024; Widder, 2024)—AI research crosses disciplinary boundaries. The cross-disciplinary challenges of AI research mirror those of other disciplines (Stokols et al., 2003; Stirling, 2014; Amoo et al., 2020; Vestal & Mesmer-Magnus, 2020; The Royal Society, 2024).

One major challenge is disciplinary silos, where knowledge is inadequately shared across disciplines (Stokols et al., 2003; Stirling, 2014; Ballantyne, 2019; Amoo et al., 2020; DiPaolo, 2022; The Royal Society, 2024). For instance, lack of knowledge sharing could be partly responsible for low attention to the distinction between explanatory and exploratory research in ML, discussed in §2.2 (Herrmann et al., 2024).

Another challenge is epistemic hierarchies—where the expertise of some disciplines is explicitly or implicitly devalued (Knorr Cetina, 1999; 2007; Simonton, 2004; Fourcade et al., 2015; Graziul et al., 2023). This can manifest as expert groups being limited to narrow input rather than contributing to broader research design decisions (Bertelsen et al., 2024).

AI researchers' focus on applying their work to other domains creates another major challenge. Insufficient domain knowledge might affect the functionality of AI tools—whether they operate as advertised (Raji et al., 2022). For instance, AI tools are deployed to make predictions about future individual-level outcomes, from pre-trial risk prediction and predictive policing to automated employment decisions (Wang et al., 2024). Yet inadequate evidence of effectiveness often fails to prevent predictive tools from being built, marketed, and deployed (Doucette et al., 2021; Cameron, 2023; Connealy et al., 2024).

The problem of disciplinary silos becomes particularly acute in AGI-oriented research due to two factors: its claims to be creating cognates or replacements of human intelligence, and its claims to expertise in many disciplines. In the former case, claims are often made while ignoring debates in cognitive science and psychology about the nature of intelligence (Summerfield, 2023; Guest & Martin, 2024; Mitchell, 2024; Van Rooij et al., 2024). In the latter case, achieving AGI is often framed in terms of being able to "replace" domain experts in various domains—which are then often taken up uncritically by the media without input from domain experts themselves (e.g., Henshall, 2024).

**Problem 3: Resource disparities.** In recent years, we have witnessed an unprecedented growth in computational resources required for model training, with requirements doubling approximately every few months (Sevilla et al.,

2022). This trend compounds existing resource disparities, as state-of-the-art AI research often relies on access to computational resources accessible to very few researchers (Yu et al., 2023; LaForge, 2024). The financial cost of these resources excludes a wide range of actors from contributing to AI research, as even top research universities have a fraction of the computational resources that many corporations use to advance AI research. Efforts are underway to address this resource disparity by supporting access to large-scale computational resources maintained by government entities (e.g., NAIRR in the United States). Yet, resource parity is an aspirational goal in response to widespread recognition that AI researchers in industry enjoy a *de facto* advantage in setting the goals of AI research due to their access to industrial scale computational resources. This structural advantage is reinforced by the use of pre-print archives to publicize AI research without peer review (Devlin et al., 2019; Rombach et al., 2022; Bubeck et al., 2023; OpenAI et al., 2024), a strategy which legitimizes this work as scientific in nature without applying traditional standards for scientific integrity (Tenopir et al., 2016; Lin et al., 2020; Soderberg et al., 2020; Rastogi et al., 2022; Kwon & Porter, 2025).

While resource disparities exist for all forms of AI research, they are particularly stark when AGI is taken as a north-star goal for the discipline, due to the orientation of current AGI efforts towards sheer computational scale and the concentration of such efforts in large tech companies.[7] Such concentration of power makes it even more important that those efforts include, rather than exclude, relevant communities and experts. That is, AGI discourse accelerates the existing trend in AI of discounting domain expertise and lived experiences in favor of models that are allegedly experts in everything.

## 3. Recommendations

We have argued that AGI discourse hinders setting well-motivated scientific and engineering goals in AI development, while being destructive to the development of AI that has social merit. We now provide three recommendations for avoiding these traps.

**Recommendation 1: Goal Specificity.** *The AI community must prioritize **highly specific** language when discussing the scientific, engineering, and societal goals of AI.*

More specific definitions of tangible scientific, engineering, and societal goals promote a shared understanding of these goals, and thus the capacity to evaluate whether these goals are well-motivated. Without such specificity, researchers, practitioners, and others can develop divergent understand-

ings of a goal and how it should be achieved. This divergence enables conceptual arbitrage on the part of AI researchers and practitioners who seek to advance their own goals, since these actors ultimately determine the details of model development and implementation. People external to AI development may then be left assuming that a system achieves a specific goal when, in fact, it does not.

Specificity can maintain sufficient flexibility for exploratory research by engaging in best practices around developing a research question. Consider the research goal "How can a mixture of experts (MoE) strategy improve performance of Whisper large-v3 in challenging speech domains?" which could cover various domains of speech or MoE strategies. A reformulated, more specific goal would be: "How can a MoE strategy, where experts are a series of models fine-tuned on short (<2s), medium (2–20s), and long utterances (>20s), help improve performance of Whisper large-v3 in a speech domain dominated by short utterances but also containing relatively long utterances?" Without additional specification, answering the first question provides little guarantee that a solution would address the specific features of the second question, at least not without substantial effort to understand how such a general solution may be applied to this specific case/domain. This example is illustrative, though based on real features of a speech domain where accuracy is essential (i.e., police radio communications, see Srivastava et al. 2024; Venkit et al. 2024).

Goal Specificity addresses the Illusion of Consensus trap (§2.1), promoting conceptual clarity as an essential part of goal-setting. Clarity also helps to avoid the Goal Lottery trap (§2.4) by making goal selection explicit. It similarly addresses the Presuming Value-Neutrality Trap (§2.3) by explicitly surfacing values tied to specific goals, and directly reduces Generality Debt (§2.5) . Finally, goal specificity addresses the underspecification issues highlighted in the Supercharging Bad Science Trap (§2.2).

**Recommendation 2: Pluralism of goals and approaches.** *Rather than a single general north-star goal (or small set of goals), the AI community should articulate **many** worthwhile scientific, engineering, and societal goals—and many possible paths to fulfilling them.*

Reaching meaningful scientific and societal consensus on the goals of a field as broad-ranging as AI is challenging. When consensus may not be viable or desirable, we recommend pluralism: allowing multiple viable conceptions of the goals of AI research. Pluralism is healthy in a society composed of individuals and institutions with divergent values. By default, the research community should be pluralistic about goals and paths to achieving them, aiming for heterogeneity instead of homogeneity (Sorensen et al.,

---

[7]Large-scale efforts also have detrimental impacts on climate, reinforcing resource disparities (Bucknall & Dori-Hacohen, 2022; Kaack et al., 2022; Luccioni et al., 2024).

2024a;b).[8]

Researchers who study the dynamics of knowledge production and problem-solving in groups have found pluralism to be beneficial (Hong & Page, 2004; Muldoon, 2013), including unique benefits ascribable to egalitarian group dynamics (Xu et al., 2022).

Pluralism in AI research can take several forms, each with distinct implications for how we approach complex problems. For example:

**Methodological pluralism** implies that different ways of approaching a problem help achieve better solutions, often an effective strategy for complex problem-solving in general (Midgley, 2000; Veit, 2020; Zhu, 2022) .

**Value pluralism**, as applied to alignment research, implies a direct connection between technical advancement and accommodation of plural values (Sorensen et al., 2024a;b).

**Algorithmic pluralism** addresses the "patterned inequality" associated with algorithmic monoculture and implies a "plurality of paths to different outcomes" must be supported to avoid reproducing existing social inequalities (e.g., embedded in data) (Jain et al., 2024) .

Each form of pluralism translates into concrete research practices, such as choice of method(s), setting of pluralistic goals, and testing algorithms to ensure known harms (i.e., algorithmic discrimination) are addressed.

To name just one example, algorithmic decision-making in hiring processes is unlikely to benefit from AGI as much as from targeted solutions to that particular setting, which account for the domain-specific nature of most positions, the different ways hiring managers evaluate candidates, and existing evidence of algorithmic discrimination in hiring. In practice, pluralism can manifest in how resources are distributed among different research approaches. For example, rather than investing most computational resources in pursuit of AGI, they could be more evenly distributed among diverse goals and approaches within AI.

Pluralism addresses the Illusion of Consensus trap (§2.1) by acknowledging the lack of consensus, and using diversity of perspectives as a tool for scientific and social progress; the Goal Lottery trap (§2.4) by reducing the chances of arbitrarily or prematurely excluding some goals from consideration; and the Exclusion trap (§2.6) by encouraging a plurality of goals and approaches.

---

[8]We're not rejecting consensus on unifying, general north-star goals as a matter of principle. In some circumstances, like coordinating collective action in response to the climate crisis, strong consensus may become crucial. But the research community should not begin from the assumption that such consensus is necessary, or that consensus is optimal from a scientific or societal perspective.

**Recommendation 3: Greater Inclusion in Goal Setting.**
*Greater inclusion of communities and disciplines in shaping the goals of AI research is beneficial to innovation.*

Inclusion supports innovation (Burt, 2004; Hewlett et al., 2013; Zhang et al., 2021; Xu et al., 2022). Identifying worthwhile goals, related use cases, and potential unintended consequences depends on engaging diverse viewpoints. Within AI research, these viewpoints must include those of end users, experts from other fields, those affected by research outcomes, and data annotators. Excluding any of these groups impoverishes the potential of AI to achieve worthwhile goals since it would discount the perspectives that define these goals as worthwhile. Including them enriches AI research.

Cross-pollination of ideas between disciplines leads to more impactful research (Dori-Hacohen et al., 2021; Shi & Evans, 2023). Such impact requires that we abandon silos of (tacit) knowledge (e.g., epistemic cultures: Knorr Cetina, 1999; 2007) and prioritize epistemic hierarchies that value non-computational research (Simonton, 2004; Fourcade et al., 2015). While technical complexity can make participation by non-experts challenging (Pierre et al., 2021), working through these challenges can surface issues experts have not anticipated (Cooper et al., 2022) and enable practical scientific contributions by integrating the insights and experiences of non-experts, or experts in other fields, into system design decisions (Delgado et al., 2023; Salavati et al., 2024).

As a topic, AGI often involves imagining AI technologies that impact the lives of everyone. Exclusion thus causes socially significant disagreements regarding the goals, processes, and actors who shape AI research and deployment. These disagreements are often overlooked or ignored by those with the power to shape the field. Inclusion is necessary to ensure that decisions about technology are sufficiently justified to institutions, communities, and individuals (Anderson, 2006; Binns, 2018; Alexandrova & Fabian, 2022; Birhane et al., 2022a; Lazar, 2022; Ovadya, 2023).

This recommendation addresses the Normalized Exclusion trap (§2.6) by treating inclusion as essential to innovation. Moreover, it addresses the Illusion of Consensus and the Presuming Value-Neutrality traps (§2.1, §2.3) by acknowledging socially significant disagreements about the value-laden goals of AI research and development and using those disagreements productively.

## 4. Alternative Views

We have argued that AGI is a poor choice as a north-star to guide AI research. We conclude by championing our position against a strong alternative view: that the traps we have identified can be addressed through a modified pursuit

of AGI. We argue that improved approaches to AGI would not go far enough.

## 4.1. Counterargument

"*AGI is a good north-star goal; to avoid the above traps, improved definitions and accounts of AGI are needed.*"

Thoughtful attempts to address shortcomings in accounts of AGI indeed exist (e.g., Adams et al., 2012; Chollet, 2019; Summerfield, 2023; Morris et al., 2024). If prior accounts of AGI are counterproductive or flawed, why not pursue new accounts that address those flaws?

As an example of an alternative view, Morris et al. (2024) arguably mitigate the Illusion of Consensus trap by disentangling the disagreements about goals, predictions, and risks that plague other accounts of AGI. Moreover, their proposal for a practical strategy analogous to Levels of Driving Automation standards (SAE International, 2021) could be viewed as mitigating the Presuming Value-Neutrality trap. Setting society-wide standards could be done in a way that explicitly centers *specific* risks that the standards address (Morris et al., 2024). In this way, the values being favored manifest in the risks that are centered by the standards. Such works showcase a potential response to traps. In arguing that AGI discourse aggravates multiple standing problems, we cannot rule out the possibility of efforts that mitigate these same problems while retaining AGI as a goal. Moreover, given that lack of agreement about how to define AGI is likely to persist, as Summerfield (2023), Morris et al. (2024), and many others believe, it would be especially implausible for us to presume to have a complete enough view of the landscape of possible conceptions of AGI to draw definitive conclusions.

## 4.2. Rebuttal

Why favor our position against this alternative?

**Reason 1: Conflict with our recommendations.** Although the definition of AGI is highly contested, a frequent motivation for embracing AGI as a north-star goal is the desire for a single, large-scale, unifying vision for the field (Summerfield, 2023). This is somewhat in tension with our recommendation of goal pluralism, which we argue is valuable for AI research. "AGI" also brings with it a notion of "general" that can be discordant with our recommendation of specificity.[9] As such, there are reasons to be wary of AGI-focused alternatives.

---

[9]Note that these are not *all things considered* (or definitive) reasons to abandon the alternative view. Rather, readers who are convinced by our arguments in favor of specificity and pluralism have, to some extent, reasons to be wary of the project of improving AGI. These are so-called *pro-tanto* (i.e., "to that extent") reasons, which can be overridden by other considerations (Alvarez, 2023).

**Reason 2: Distinguishing hype from reality.** Another reason to favor our position is the AI research community's responsibility to help distinguish hype from reality. We believe that our community must provide trusted, evidence-based answers to increasingly complex questions about AI technologies, their goals, and their impacts. In the current moment, the hyped terminology of AGI undermines this responsibility.

No matter how well or poorly defined, AGI has acquired a cultural significance that exacerbates the challenge of distinguishing hype from reality. "Intelligence" and "generality" hold the promise of being beneficial for countless needs and contexts (§2.3, §2.5). No matter how cautious the research community attempts to be, the cultural associations of these terms risk stoking the flames of unscientific thinking about AI. This enables various parties to loosely project utopian or dystopian characteristics onto AGI in ways that support their calls for more power and resources.

**Reason 3: Benefiting humans as the goal of technology.** If the AI community nevertheless wants an overarching goal to strive towards, the goal should be the support and benefit of human beings. The goals of technology are shaped by *people*. Evidence-based approaches to examining whether technology effectively meets the needs of people—be they "users", "consumers", "patients", "scholars", or a myriad of business and social monikers—are well-established. In a quest to achieve AGI, communities often lose sight of the needs of people as a goal, in favor of focusing on just the technology.

There is another, more ambitious reason to work towards consensus on supporting and benefiting human beings as a goal. We have noted the role of socially significant disagreements about the goals of technology in our third recommendation of inclusion. Processes ensuring that technology benefits humans have the potential to provide *collectively legitimate* responses to such disagreements. This could occur through processes that embrace democratic ideals: such as universal inclusion in interrogation, deliberation, and dissent about the "common good" and "public interest", while enacting strong accountability to participants as "rights-holders" (Anderson, 2006; Putnam, 2011; Binns, 2018; Gabriel, 2020; Birhane et al., 2022a; Lazar & Nelson, 2023; Ovadya, 2023; Blili-Hamelin et al., 2024). Aiming for collective legitimacy amounts to requiring politically and socially effective forms of consensus.

In sum, we urge communities to **stop treating "AGI" as the north-star goal of AI research**.

## Acknowledgments

Borhane Blili-Hamelin was funded in part through a Magic Grant from the Brown Institute for Media Innovation.

Christopher Graziul was supported by the National Institute of Minority Health and Health Disparities of the National Institutes of Health under award number R01MD015064. This material is based upon work supported in part by the NSF Program on Fairness in AI in Collaboration with Amazon under Award IIS-2147305. Any opinions, findings, and conclusions or recommendations expressed in this material are those of the author(s) and do not necessarily reflect the views of the National Science Foundation, National Institutes of Health, the Brown Institute for Media Innovation, or Amazon.

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

## A. Definitions of AGI and Related Concepts

**Table 1** below presents illustrative definitions of AGI and usefully related concepts. We agree with Morris et al. (2024)'s proposal to broaden discussions of AGI definitions to include accounts that avoid the term "AGI" yet address similar goals of achieving human-level intelligence. For example, while OpenAI's influential definition (OpenAI, 2018) focuses on outperforming humans at economically valuable work, sharing key parallels with Nilsson (2005). Yet it notably differs from Chollet et al. (2024); Summerfield (2023); Morris et al. (2024); Chollet (2019); Goertzel (2014) and others by not explicitly emphasizing generality.

Following Blili-Hamelin et al. (2024), we believe discussions of AGI definition should include approaches that challenge AGI's central premises. Below, we include Weizenbaum (1976) and Attard-Frost (2023). Including these critical accounts enables noticing a surprising similarity with Summerfield (2023)'s reconceptualization of AGI through the lens of natural intelligence: all three accounts favor a strong form of contextualism and pluralism about what intelligence means.

## B. AGI as a North-Star Goal

This paper argues against AGI serving as a north-star goal of AI research. When we talk about AGI being treated as north-star goal, we are not claiming that the majority of AI researchers are explicitly working in pursuit of this goal. In fact, many AI researchers may, like us, doubt or reject this goal (Salvaggio, 2025; Francesca Rossi et al., 2025). Rather, we claim that influential researchers and executives hold this view, enough so for it to deserve scrutiny. These dominant voices are further amplified by the publicity that discussions of AGI generate, including by members of the press (Klein, 2025), and by government commissions (Bommasani et al., 2025). As such, AGI has come to permeate both community incentives and cultural norms. In this context, interrogating its role and influence in the AI research community matters. Below we provide a small sample of quotes and resources illustrating this effect.

**OpenAI** The mission statement of OpenAI is "...to ensure that [AGI] benefits all of humanity." Its website (OpenAI, 2025a) states that "we are building safe and beneficial AGI, but will also consider our mission fulfilled if our work aids others to achieve this outcome." This goal directly influences both the company's direct work (OpenAI, 2025b) and the work that it funds (OpenAI, 2025c).

**Google DeepMind** The vision statement (DeepMind, 2025) of Google DeepMind states that "[AGI] has the potential to drive one of the greatest transformations in history." In a recent briefing (Browne, 2025), Demis Hassabis stated that, though current systems still have limitations, over the next 5–10 years "a lot of those capabilities will start coming to the fore and we'll start moving towards what we call [AGI]." A position paper (Morris et al., 2024) by Google DeepMind authors last year defines concrete goals in pursuit of AGI.

**Anthropic** In the essay "Machines of Loving Grace," Anthropic CEO Dario Amodei argues how the world could be shaped positively by "Powerful AI" aligned with "AGI" goals (Amodei, 2024). Amodei recently told CNBC that AI that is "better than almost all humans at almost all tasks" can emerge shortly (Browne, 2025). Anthropic's official recommendations (Anthropic, 2025) to OSTP for the U.S. AI Action Plan state that "we expect powerful AI systems will emerge [with] intellectual capabilities matching or exceeding that of Nobel Prize winners."

**Other Influential Executives & Researchers** In "Sparks of AGI" (Bubeck et al., 2023), researchers at Microsoft argue that GPT-4 "could reasonably be viewed as an early (yet still incomplete) version of [AGI]." In announcing xAI, Elon Musk stated (Kerner, 2023) that "the overarching goal of xAI is to build a good AGI." Speaking to TIME (Perrigo, 2024), Yann LeCunn explained that he refers to "what people call 'AGI'" as "human-level intelligence," and noted that "the mission of FAIR [Meta's Fundamental AI Research team] is human-level intelligence." Geoff Hinton stated to Forbes (Maymin, 2023) that he "is certain we will have AGI soon, and biological humans will be relegated to be the second-smartest species on the planet."

Table 1: Sample of proposed definitions of AGI and related concepts.

Weizenbaum (1976). "Intelligence is a meaningless concept in and of itself. It requires a frame of reference, a specification of a domain of thought and action, in order to make it meaningful. [. . . ] [T]hese domains are themselves not measurable." Argues that any argument that calls for the conclusion or denial that "machines may surpass us in general intelligence" is "ill-framed and therefore sterile" due to "our inability to compute an upper bound on machine intelligence." We follow Blili-Hamelin et al. (2024) in considering this critical account relevant to debates about how to conceive AGI.

Searle (1980). "according to strong AI, the computer is not merely a tool in the study of the mind; rather, the appropriately programmed computer really is a mind, in the sense that computers given the right programs can be literally said to understand and have other cognitive states."

Gubrud (1997) "By advanced artificial general intelligence, I mean AI systems that rival or surpass the human brain in complexity and speed, that can acquire, manipulate and reason with general knowledge, and that are usable in essentially any phase of industrial or military operations where a human intelligence would otherwise be needed. Such systems may be modeled on the human brain, but they do not necessarily have to be, and they do not have to be "conscious" or possess any other competence that is not strictly relevant to their application. What matters is that such systems can be used to replace human brains in tasks ranging from organizing and running a mine or a factory to piloting an airplane, analyzing intelligence data or planning a battle."

Nilsson (2005). "achieving real human-level artificial intelligence would necessarily imply that most of the tasks that humans perform for pay could be automated. Rather than work toward this goal of automation by building special-purpose systems, I argue for the development of general-purpose, educable systems that can learn and be taught to perform any of the thousands of jobs that humans can perform."

Fast Company (2010) "*Wozniak: Could a Computer Make a Cup of Coffee?*" tasks the machine to go into an "average" American home, find ingredients, and make a cup of coffee. This requires embodied AI systems. Wozniak's test has since been included in discussions of AGI (Goertzel et al., 2012).

Chalmers (2010). "AI is artificial intelligence of human level or greater (that is, at least as intelligent as an average human). Let us say that AI+ is artificial intelligence of greater than human level (that is, more intelligent than the most intelligent human). Let us say that AI++ (or superintelligence) is AI of far greater than human level (say, at least as far beyond the most intelligent human as the most intelligent human is beyond a mouse)."

Goertzel et al. (2012). Propose an architecture for human-like general intelligence that integrates slightly modified versions of previously existing architectures, emphasizing the commonalities across different approaches.

Bostrom (2014). "We can tentatively define a superintelligence as *any intellect that greatly exceeds the cognitive performance of humans in virtually all domains of interest.*"

Goertzel (2014). "roughly speaking, an AGI system is a synthetic intelligence that has a general scope and is good at generalization across various goals and contexts."

Smart (2015). "a strong AI system would be an entirely autonomous computer system in no way controlled or influenced by human operators. It could successfully adapt to its environment or even be part of its environment, making intelligent decisions, and for all intents and purposes interacting with humans naturally. It would have vastly superior memory and computational abilities but would also be able to reason and act accordingly. What all of this boils down to is that a strong AI would have to be conscious."

OpenAI (2018). "OpenAI's mission is to ensure that artificial general intelligence (AGI)—by which we mean highly autonomous systems that outperform humans at most economically valuable work—benefits all of humanity." December 2024 reporting suggests that OpenAI and Microsoft "signed an agreement last year stating OpenAI has only achieved AGI when it develops AI systems that can generate at least $100 billion in profits" (Zeff, 2025). If true, this is a significant departure from their former definition.

Chollet (2019); Chollet et al. (2024). Defines AGI as "a system capable of efficiently acquiring new skills and solving novel problems for which it was neither explicitly designed nor trained." In 2019, introduced an as yet (January 2025) unsolved benchmark for incentivizing progress towards AGI thus defined. Proposes that "it's still feasible to create unsaturated, interesting benchmarks that are easy for humans, yet impossible for AI – without involving specialist knowledge. We will have AGI when creating such evals becomes outright impossible" (Chollet, 2024b).

Hernández-Orallo et al. (2021). "independently of its overall capability, *an agent can only be called fully general if it covers all tasks up to an equivalent level of difficulty, determined by the resources that are needed for them.*" Introduces two individual-specific (be them humans, other animals or AI systems) measures that, together, decouple the concept of general intelligence: (i) 'generality', which refers to "the *distribution* of the tasks the agent can solve," and (ii) 'capability', which indicates "how far, on average, an agent can reach in terms of task difficulty."

Marcus (2022). Defines AGI as "a shorthand for any intelligence. . . that is flexible and general, with resourcefulness and reliability comparable to (or beyond) human intelligence." Calls for the need to operationalize the definition as a single system that can succeed in at least 3 of 5 proposed tasks, including Wozniak's coffee cup benchmark (Fast Company, 2010).

Morris et al. (2024). Proposes a practical strategy analogous to Levels of Driving Automation standards (SAE International, 2021). Describes a graded set of levels of achievement of target characteristics that can each be associated with tangible "metrics", the introduction of "risks", and changes in "Human-AI Interaction paradigm" (Morris et al., 2024). The framework targets 2 characteristics: levels of performance (which they define as "the depth of an AI system's capabilities, i.e., how it compares to human-level performance for a given task"), and levels of generality (defined as "breadth of an AI system's capabilities, i.e., the range of tasks for which an AI system reaches a target performance threshold").

Summerfield (2023) We consider this account to endorse a strong form of pluralism about intelligence: natural intelligence takes many different shapes across cultures and species, serving many goals and functions that are irreducibly shaped by "the internal model by which an animal understands the world", which itself "depends on its local environment, its embodied form, its desires and goals, and its interactions with conspecifics." The same should be expected for "strong AI" or "AGI". However, building on Dreyfus & Dreyfus (1986), Summerfield argues we have practical reasons to constrain the forms AI takes. The goal of AI is "to help humans in their endeavours." To that end, "if we want to build AI systems that exhibit human-like intelligence, with whom we can interact in pursuit of human-centred goals, these agents will need to think in ways that make sense to us."[10]

Attard-Frost (2023) Defines human and artificial intelligence as "value-dependent cognitive performance", and "centres interdependencies between agents, their environments, and their measurers in collectively constructing and measuring context-specific performances of intelligent action." Although this account is not presented as a conception of AGI, Blili-Hamelin et al. (2024) argue that the account is relevant to the topic.

Suleyman & Bhaskar (2023) "Artificial intelligence (AI) is the science of teaching machines to learn humanlike capabilities. Artificial general intelligence (AGI) is the point at which an AI can perform all human cognitive skills better than the smartest humans."

Agüera y Arcas & Norvig (2023). "'General intelligence' must be thought of in terms of a multidimensional scorecard, not a single yes/no proposition." Dimensions discussed include topics, tasks, modalities, languages, and instructability.

---

[10]"[W]e are building AI to make the world a better place. But if we want AI to be useful to people, it will need to share our umwelt. If we build an AI that sees the world in a radically different way to us, its behaviour and mental states will be unintelligible. Such an agent will be at best unreliable and at worst unsafe." (Summerfield, 2023)

## Author Contributions

We follow the CRediT recommendations and taxonomy provided by Allen et al. (2019) to determine and outline author contributions.[11]

- Borhane Blili-Hamelin: Conceptualization (Formulation & Evolution), Investigation, Methodology (Development), Project administration, Supervision (Oversight & Leadership), Writing (Initial draft, Submitted draft, Review & Editing).

- Christopher Graziul: Conceptualization (Formulation & Evolution), Investigation, Methodology (Development), Project administration, Supervision (Oversight & Leadership), Writing (Initial draft, Submitted draft, Review & Editing).

- Leif Hancox-Li: Conceptualization (Formulation & Ideas), Investigation, Methodology (Development), Supervision (Mentorship), Writing (Initial draft, Submitted draft, Review & Editing).

- Hananel Hazan: Conceptualization (Ideas & Evolution), Investigation, Methodology (Development), Project administration, Writing (Initial draft, Review & Editing, LaTeX).

- El-Mahdi El-Mhamdi: Conceptualization (Evolution & Ideas), Writing (Initial draft, Review & Editing).

- Avijit Ghosh: Conceptualization (Ideas), Writing (Submitted draft [Traps section], Review & Editing).

- Katherine Heller: Conceptualization (Formulation & Evolution), Supervision (Mentorship), Writing (Initial draft, Submitted draft, Review & Editing).

- Jacob Metcalf: Conceptualization (Formulation & Evolution), Writing (Submitted draft, Review & Editing).

- Fabricio Murai: Conceptualization (Ideas), Writing (Submitted draft [Table of AGI definitions], Review & Editing, LaTeX).

- Eryk Salvaggio: Conceptualization (Evolution & Ideas), Writing (Submitted draft [Introduction, Traps section], Review & Editing).

- Andrew Smart: Conceptualization (Formulation & Evolution), Writing (Initial draft, Review & Editing).

- Todd Snider: Conceptualization (Ideas), Methodology (Implementation), Writing (Initial draft, Review & Editing, LaTeX).

- Mariame Tighanimine: Conceptualization (Evolution & Ideas), Writing (Initial draft, Review & Editing).

- Talia Ringer: Conceptualization (Formulation, & Ideas), Project administration, Supervision (Oversight & Leadership), Writing (Initial draft, Review & Editing).

- Margaret Mitchell: Conceptualization (Ideas & Evolution), Investigation, Methodology (Development), Supervision (Oversight & Mentorship), Writing (Initial draft, Submitted draft [all sections], Review & Editing, Rebuttal).

- Shiri Dori-Hacohen: Conceptualization (Ideas & Evolution), Investigation, Methodology, Project administration, Supervision (Oversight & Leadership), Writing (Initial draft, Submitted draft [all sections], Review & Editing).

---

[11]The initial ICML submission was created by Borhane Blili-Hamelin, Leif Hancox-Li, and Christopher Graziul, leveraging extensive work and writing from the larger project. All contributors then worked together on refining this submission.

