# OpenReview forum: "Position: Stop treating `AGI' as the north-star goal of AI research"
_ICML.cc/2025/Position_Paper_Track — ICML 2025 Position Paper Track poster_

### Official Review · Reviewer_XPzd · 2025-02-17

**Significance:** 3
**Argument Clarity:** 3
**Rating:** 4
**Confidence:** 4

**Questions:**

Can you provide specific examples of research projects or initiatives that were negatively impacted by AGI discourse in ways that illustrate your identified "traps"? This would strengthen the causal claims made throughout the paper.

How do you envision implementing the "specificity" recommendation while maintaining sufficient flexibility for exploratory research? Could you provide concrete examples of how research goals should be reformulated?

Your framework presents six interrelated traps - could you develop a more formal theoretical model showing how these traps interact and reinforce each other? This would help readers understand whether the set is complete and properly structured.

Given the historical precedent of debates between general and narrow AI approaches, how does your analysis differ from or build upon similar arguments made in previous eras of AI development?

**Discussion Potential:**

4

**Paper Summary:**

The paper presents a clear and well-structured argument against treating AGI as the north-star goal of AI research. The position is explicitly stated in the title and abstract, and the paper systematically develops its case through six "traps" that the authors argue are exacerbated by AGI discourse.

**Position:**

Yes

**Position In Title:**

Yes

**Related Work:**

3

**Strengths And Weaknesses:**

The manuscript demonstrates strong engagement with recent literature, citing work through 2024 and integrating perspectives from multiple disciplines. The authors effectively synthesize research from computer science, philosophy of science, and science and technology studies to support their position.
The paper's organization is logical, with each "trap" building on previous arguments. The authors address potential counterarguments in Section 4, demonstrating scholarly rigor in considering alternative viewpoints.
Significant Concerns:

Theoretical Framework Development: While the paper identifies six "traps," the theoretical relationships between these concepts could be more rigorously developed. For example, the authors note that the traps are "highly interrelated" but don't provide a formal framework for understanding these relationships. This makes it difficult to evaluate whether the set of traps is complete or properly structured.

Validation Approach: The paper relies heavily on citation and argument rather than providing empirical evidence for its claims. While this is acceptable for a position paper, the authors could strengthen their case by providing more concrete examples of how AGI discourse has directly led to the identified problems. Many claims about AGI's negative effects remain at a theoretical level.

Implementation Guidance: While the paper provides three recommendations in Section 3, these remain somewhat abstract. For instance, the recommendation for "Goal Specificity" could benefit from concrete examples of how research goals should be reformulated. The paper could provide more practical guidance for implementing its recommendations.

Literature Gap: Though the paper engages well with recent work, it could better address historical precedents of similar debates in AI. The field has seen previous discussions about general versus narrow AI, and this historical context could strengthen the argument.

Specific Issues:

1. The paper claims AGI discourse "undermines the ability of researchers to choose effective goals" (Abstract) but could provide more direct evidence of this causal relationship.
2. The "Illusion of Consensus" trap discussion could benefit from quantitative analysis of how AGI is defined differently across the field.
3. The "Normalized Exclusion" section makes strong claims about AGI discourse intensifying exclusion but could provide more concrete examples.

**Support:**

3

---

> ### Author Rebuttal · Authors · 2025-04-01
>
> We thank the reviewer for their thorough engagement with our work and detailed evaluation of its logic and supporting evidence. We will revise accordingly:
>
> Q1: Direct evidence that AGI discourse undermines researchers’ goal-setting comes in two main forms, we will clarify (see also “Goal Lottery”):
>
> a. Public discussion of AGI, its goals, and its theoretical benefits indicates that resources and economic incentives align to support its pursuit as a priority over other forms of AI research.
>
> b. Claiming advances toward achieving AGI is viewed a-priori as demonstrating its practical value when:
>
>   - such value is better demonstrated by other forms of AI research, and
>
>   - there is no AGI-based system for which similar value can be demonstrated.
>
> Assuming AGI will be transformative has led to substantial resource investments in its pursuit (see Reviewer XaZq rebuttal). Since AGI doesn't currently exist, this major resource reallocation is based on, at best, direct evidence that other AI-based systems are valuable in myriad settings. Thus, AGI only exists as an abstract potential being pursued over existing forms/goals of AI research.
>
> Q2: We understand and appreciate this concern. In a-c of the Reviewer u8hJ rebuttal, we elaborate specific actionable recommendations around pluralism, and will add similar clarity about the implementation of specificity:
>
>  - (a) Specificity can maintain sufficient flexibility for exploratory research by engaging in best practices around developing a research question. For example, a research goal may be “How can a mixture-of-experts strategy improve performance of Whisper large-v3 in challenging speech domains?” yet focus on a series of domains or a series of MoE strategies.
>
> - (b) A reformulated goal would be: “How can a mixture-of experts, where experts are a series of models fine-tuned on short (<3s), medium (3-30s), and long utterances (>30s), help improve performance of Whisper large-v3 in a speech domain dominated by short utterances but also containing relatively long utterances?” This example is illustrative but based on real features of a speech domain where accuracy is essential (i.e., police radio communications).
>
> Without additional specification of the research question, (a) may be achievable but there is little to no guarantee answering this question will support (b) without substantial effort to understand how the general solution applies to this specific case/domain. Further, focus on (b) is preferred given domain-specific considerations which may or may not be aligned with efforts to pursue (a).
>
> Q3: We appreciate this observation since it parallels how our thinking about the traps evolved over time: Rather than assuming a set of structured relationships (i.e., a formal theoretical model), we opted to focus on the most self-consistent descriptions of each trap that best align with what critics and researchers have said about each.
>
>   - This is an inductive approach identifying how AGI amplifies known issues within AI research, which is our position. We believe a formal model may exist relating these traps to each other in structured ways (e.g., through political economy), but it is beyond the scope of our paper to suggest such formal relationships. Also, we do not claim our list of traps is complete.
>
>   - We focused on traps that are distinct and well-supported by research to improve the clarity of our position given its abstractness (e.g., arguing against development of a technology that does not exist). We hypothesize that such structure could emerge as research advances and the relationships between traps become clearer.
>
> Q4:
>
>   - Our analysis builds on Morris et al. (2024) “Levels of AGI”, specifically responds to its purpose: “Goals: Achieving human-level ‘intelligence’ is an implicit or explicit north-star goal for many in our field, from the 1956 Dartmouth AI Conference (McCarthy et al., 1955)...today’s leading AI research firms...mission statements include goals such as ‘ensure transformative AI helps people and society’ (Anthropic, 2023a) and ‘ensure that artificial general intelligence benefits all of humanity’ (OpenAI, 2023).” We build on their argument by engaging with contemporary concerns about AI research, but we differ by questioning the wisdom of AGI as a “north-star goal” in AI research.
>
>   - We also build on the March 2025 report from the AAAI 2025 Presidential Panel on the Future of AI Research. The section concerning AGI is chaired by Horvitz and Russell (2025). It notes the historic centrality of human-level AI as a research goal. This report also indicates most AI researchers surveyed (76%) would prefer systems designed to prioritize “risk-benefit profile over the direct pursuit of AGI (23%),” and practical and ethical questions remain about the feasibility/desirability of AGI. We differ by focusing explicitly on the link between harms of AI research and AGI as a catalyst for these harms given the traps we enumerate.

---

> > ### Comment · Reviewer_XPzd · 2025-04-08
> >
> > Thank You to the authors for addressing all of my raised comments and questions. I maintain my 4 'accept

---

### Official Review · Reviewer_k138 · 2025-03-12

**Significance:** 4
**Argument Clarity:** 3
**Rating:** 3
**Confidence:** 4

**Questions:**

How can the field orient away from AGI as the ‘north-star’ goal, if the obstacles to productive goal setting stem from having AGI as a goal?

**Discussion Potential:**

4

**Paper Summary:**

This paper argues that attaining artificial general intelligence (AGI) should not be the ultimate ‘north-star’ goal of AI research. It presents six ‘traps’ by which focusing on attaining AGI negatively impacts goal setting in AI. The paper recommends prioritising specificity and pluralism as a way of avoiding these traps.

## Update after rebuttal
I would like to thank the authors for their engagement in constructive discussion.
Based on this discussion, my predominant concern regarding the circularity of argument has largely been addressed through the authors stating that they *‘do not pretend that the traps cause AGI to become a goal’* and that *‘[r]ather than “causes”, our traps are better understood as diagnosing problems in goal setting that, if our arguments are correct, are **normatively** worth addressing.’* Based on this, I am upgrading my recommendation to ‘weak accept’ on the assumption that such clarifications are included in a revised version of the paper.

**Position:**

Yes

**Position In Title:**

Yes

**Related Work:**

4

**Strengths And Weaknesses:**

**Strengths**
- The paper is very well-written and easy to read.
- The paper is well-cited and draws extensively on relevant literature throughout.
- The paper addresses an incredibly important and timely topic, and should be the focus of in-depth discussion across the ICML community. For this reason, I am glad to see that this paper has been written.

**Weaknesses**
- My main criticism of the paper is that the overall logic of the argument, as presented, is somewhat confused. The paper seems to simultaneously argue that i) focusing on AGI leads research to fall into the identified traps; and ii) the traps are *‘obstacles to productive goal setting’* – and can therefore presumably lead to AGI being a misplaced goal of the field. (In some places this is somewhat obscured by talking about *‘AGI narratives’* rather than goals, though the distinction here is not made explicit.) This circularity leads to a curious catch-22 scenario in the recommendations whereby the authors imply that in order to avoid the traps (and therefore be able to set good goals), the field should have more specific, pluralistic, and inclusive goals. While in many ways this paper is great, this fundamental confusion in the argued position feels like a critical weakness to me. (Note that this is not entirely reflected in my scores below, as each line of argument is clear and well-supported. But taken as a whole, the argument is confused.)
- A similar situation occurs in your rebuttal to the alternative view. The first reason given against the alternative view is that it conflicts with your recommendations. But since your recommendations are based on the advocated position, this is in effect just saying that one shouldn’t accept the alternative view because it contradicts your view.
- (Minor) The reference to external validity at the top of page 3 feels like it would be better described as referring to construct validity.
- (Minor) Both (Devlin et al., 2019) and (Vaswani et al., 2023) [sic] underwent peer review, and so should not be cited as examples of *‘the use of pre-print archives to publicize AI research without peer review’* (unless ‘without’ is interpreted as ‘before’ – in which case this should be clarified in the text).

**Support:**

2

---

> ### Author Rebuttal · Authors · 2025-04-01
>
> We appreciate the anonymous reviewer’s constructive criticisms; we acknowledge their concern that the logic underlying our overall argument should be improved for clarity. However, we note that we had some difficulty understanding the precise meaning of the reviewer’s question. In what follows, we will explain our best understanding, and reply based on this understanding; however, if we misunderstood the question, then we welcome further clarification, in order to better address the concern.
>
> First we will share our understanding of the reviewer’s concern: If people pursue AGI as a goal, and we believe that pursuing AGI leads to these traps, then there is no way for AGI researchers to avoid these traps. Thus, people cannot escape these traps if they're already working on AGI.
>
> While we can appreciate this concern, we contend that our argument is not, in fact, circular: We establish that each pattern of behavior (“trap”) is harmful in itself, using relevant research, prior to elaborating how AGI exacerbates it. Obstacles to productive goal setting thus stem from each trap, which already arise in AI research but are worsened when AGI is set as the north-star goal of the field.
>
> To highlight this argumentative strategy, here is a bulleted list of each trap and where we talk about (a) harms and (b) the amplifying effects of AGI:
>
> T1: Illusion of Consensus
>
>   - (a) Harms: §2.1 (¶3); §3 (R1, ¶1)
>
>   - (b) Effects of AGI: §2.1 (¶1,¶2, ¶4)
>
> T2: Supercharging Bad Science
>
>   - (a) Harms: §2.2 (P1, ¶2-4), (P2,  ¶1)
>
>   - (b) Effects of AGI: §2.2 (P1, ¶1), (P2,  ¶1-2), (P3)
>
> T3: Presuming Value-Neutrality
>
>   - (a) Harms: §2.2 (¶1, ¶3); §3 (R1, ¶2)
>
>   - (b) Effects of AGI: §2.2 (¶2, ¶4)
>
> T4: Goal Lottery
>
>   - (a) Harms: §2.4 (¶1, ¶3-4)
>
>   - (b) Effects of AGI: §2.4 (¶2, ¶4); §3 (R2, ¶2-3)
>
> T5: Generality Debt
>
>   - (a) Harms: §2.5 (¶2-6); §3 (R1, ¶1)
>
>   - (b) Effects of AGI: §2.5 (¶1, ¶4)
>
> T6: Normalized Exclusion
>
>   - (a) Harms: §2.6 (P1 ¶1, P2 ¶1-3, P3 ¶1)
>
>   - (b) Effects of AGI: §2.6 (P1 ¶2, P2 ¶4, P3 ¶2)
>
>
> By highlighting where we identify (a) harms and (b) amplifying effects of AGI linked to each trap, we recognized that a few simple edits and reorganization of our existing content may make our overall argument much clearer. These changes will make our work more accessible to the reader (and thus, hopefully, more valuable), so we again thank the reviewer for their insightful critique.
>
> As noted above, it is possible we misinterpreted the reviewer’s concern. In that case, we would welcome further clarification on what the reviewer is referring to when discussing “circularity” and “catch-22”, in order for us to best  address the original concern.
>
> We believe the above also partially addresses the reviewer’s point about our position being tautological (“...your recommendations are based on the advocated position, this is in effect just saying that one shouldn’t accept the alternative view because it contradicts your view.”). Additionally, speaking directly to this point:
>
> Our traps reflect harms tied to AI research more generally, but we contend that this does not  lead to a contradiction; we fully agree that our recommendations are indeed applicable outside AGI-oriented research. That is, better definitions of AGI cannot address lack of specificity nor accommodate pluralistic approaches without deprioritizing generality (i.e., the “G” in AGI).
>
> - This links to our argument about AGI amplifying these harms: These recommendations are at odds with how AGI has been imagined, discussed, and pursued as a goal to date.
>
> - A different alternative view we considered is that only a minority of researchers pursue AGI, so it is not an orienting perspective in contemporary AI research and development. We address this view in our rebuttal to Reviewer XaZg.

---

> > ### Comment · Reviewer_k138 · 2025-04-03
> >
> > I would like to thank the authors for their detailed response to my review, and I apologise that my concerns were not stated as clearly as they could be. I will try to clarify them below.
> >
> > In your rebuttal you summarise your understanding of my concern as:
> > >*“If people pursue AGI as a goal, and we believe that pursuing AGI leads to these traps, then there is no way for AGI researchers to avoid these traps. Thus, people cannot escape these traps if they're already working on AGI.”*
> >
> > While this is true, it does not accurately represent my concern when considered in isolation.
> >
> > My concern stems from your arguing additionally that these traps *‘hinder the research community’s ability to set worthwhile goals.’* (Sec. 2, Para. 1) Taking these two things together implies not only that *‘people cannot escape these traps if they’re already working on AGI’* (as you point out above) but furthermore that **the ability to set worthwhile goals is hindered by having AGI as a goal** (via falling into these traps). If one agrees that AGI is not a worthwhile goal (due to its exacerbating the traps), then this implies that **the ability to set worthwhile goals is hindered by having a goal that is not worthwhile** (namely, pursuing AGI). It is this last implication that I see as being circular.
> >
> > Put another way, you seem to simultaneously argue 1) that:
> > - Having AGI as a goal exacerbates the traps (Sec. 2, Para. 1)
> > - Falling into these traps is harmful (Sec. 2)
> > - Therefore, AGI is a suboptimal goal. (Conclusion, Overall position)
> >
> > (In this case, ‘AGI goal’ causes ‘Traps’);
> >
> > And 2) that:
> > - The traps cause suboptimal goal-setting. (Sec. 2, Para. 1)
> > - AGI is a suboptimal goal. (Overall position)
> > - Therefore, the traps cause a harmful focus on achieving AGI. (Conclusion)
> >
> > (In this case, ‘Traps’ causes ‘AGI goal’)
> >
> > So in the end we have that the causality flows both ways, and that ‘falling into the traps’ and ‘having AGI as a goal’ are mutually reinforcing.
> >
> >
> >
> > With regards to my concern about the first alternative view:
> >
> > You argue in §3 for goal specificity, goal pluralism, and inclusion in goal setting as *‘recommendations for avoiding these traps.’* (Sec. 3, Para. 1) I.e. These recommendations are portrayed as being instrumentally valuable for avoiding the traps, rather than being desirable ends in and of themselves.
> >
> > You then rebut the alternative view that advocates for *‘improved definitions and accounts of AGI’* by saying that this is *‘at odds’* with ‘specificity of goals and pluralism of goals.’ (Sec. 4, Reason 1) But, given the instrumental value of goal specificity and pluralism, this is prima facie no reason to abandon the alternative view: If *‘improved definitions and accounts of AGI’* are at least as instrumentally valuable for avoiding the traps as goal specificity and pluralism, then they should be the favoured approach, regardless of whether the two approaches are at odds with each other. But no argument is given for why goal specificity and pluralism are better at avoiding the traps than improved definitions and accounts of AGI. Instead, the rebuttal implies that goal specificity and pluralism are inherent goods, and thus any alternative view that is at odds with them should be rejected. But, as noted above, section 3 argues for them by pointing out their instrumental, rather than inherent, value.
> >
> > I hope that this comment has clarified my concerns, and look forward to the authors’ response.

---

> > > ### Author Response · Authors · 2025-04-08
> > >
> > > We sincerely thank Reviewer k138 for taking the time to clarify their concerns further. We now better understand that the reviewer perceives circularity in our argument arising from the mutual reinforcement between setting AGI as a goal and falling into the identified traps. We appreciate this opportunity to clarify our reasoning.
> > >
> > > First, let’s take a step back to ground our points. A key overarching theme connects the traps we identify and our rejection of the alternative view: the need for the research community to help distinguish hype from reality. In §4.2, reason 2, we highlight that this theme underpins our skepticism towards improved definitions of AGI as the best path forward. We worry that, no matter how cautious researchers try to be, the cultural associations of AGI (as well as notions of intelligence and generality) risk stoking the ﬂames of unscientiﬁc thinking about AI. A related concern appears, for instance, in our discussion of bad science (§2.2), where we emphasize the importance of sound epistemic practices and scientific validity.
> > >
> > > Regarding whether AGI as a goal causes the traps or vice versa. There is a conceptual reason why we do not pretend that the traps cause AGI to become a goal. As argued by prior work (Morris et al., Blili-Hamelin et al), there is no agreed-upon conception of AGI. Taking this claim seriously, to us, makes the project of pinpointing what “causes” AGI to become a “goal” very perilous. Rather than “causes”, our traps are better understood as diagnosing problems in goal setting that, if our arguments are correct, are **normatively** worth addressing.
> > >
> > > An example may help here:  AGI is an example of an underspecified goal—the kind of trap we describe in §2.2 problem 1—but our paper’s “bad science” examples involve far more modest goals such as “language understanding” or “imagination.” Sub‑optimal goals can arise for many other reasons (fraud, for instance), so we do not argue that this trap explains why AGI, in particular, gets chosen. Future research may reveal such a causal link, but demonstrating it is outside our paper’s scope. We rather mean to highlight the normative importance of avoiding underspecified goals in AI research, including but not limited to AGI.
> > >
> > > Regarding §4.2, reason 1, we appreciate the reviewer’s invitation to clarify this argument. **We simply meant to flag that readers who are convinced by our arguments in favor of specificity and pluralism have, to some extent, reasons to be wary of the project of improving AGI.** However, these are not definitive (all things considered) reasons. In philosophical terms, these are ‘pro tanto’ (’to that extent’) reasons: they can be overridden by other considerations.
> > >
> > > For instance, at this point in the paper, we have argued that specificity is helpful for addressing traps §2.1, §2.2, §2.3, §2.4, and §2.5. Variants of AGI that put strong emphasis on generality are, arguably, difficult to reconcile with the need for specificity. Therefore, readers who are convinced by our argument in favor of specificity have reasons to be somewhat wary of variants of AGI that put strong emphasis on generality.  Similarly, we see pluralism about goals as at odds with the project of pursuing “north-star” (i.e., orienting) goals more broadly. (See L355-358: “Rather than a single general north-star goal [...], the AI community should articulate many worthwhile scientiﬁc, engineering, and societal goals—and many possible paths to fulﬁlling them.”). Readers convinced by that argument have, to some extent, reasons to be skeptical of the alternative view. We will make this point more clearly in revising §4.2, reason 1.
> > >
> > > We appreciate the distinction you are drawing here: "the rebuttal implies that goal specificity and pluralism are inherent goods, and thus any alternative view that is at odds with them should be rejected. But, as noted above, §3 argues for them by pointing out their instrumental, rather than inherent, value."  This may be a misunderstanding due to the rebuttal’s brevity. We do not believe that any view that is at odds with specificity and pluralism should be rejected. As clarified above, reason 1 at best provides ‘pro tanto’ reasons to favor our view. In the previous rebuttal, we took ourselves to have established specificity and pluralism as instrumentally valuable for, e.g., good science; and in the rebuttal refer to them assuming this additional background is given. We hope further clarifying argument 4.2, reason 1 in the paper will help address this concern.

---

### Official Review · Reviewer_XaZq · 2025-03-14

**Significance:** 2
**Argument Clarity:** 2
**Rating:** 1
**Confidence:** 4

**Questions:**

None.

**Discussion Potential:**

2

**Paper Summary:**

The paper urges the ICML community to abandon the position of "treating AGI as the north-star goal of AI research". To support their position, the paper lists a number of issues in current AGI works, including
- definition ambiguity of AGI
- weakness in AGI benchmarks
- issues of AGI research paradigm viewed from a science/hypothesis-testing perspective
- SOTA-chasing tendency
- negative social impacts
- not being inclusive enough

The paper then advocates the AI community to set specific, plural, and inclusive goals for their research. Finally, the paper argues that even (potential) improvement of AGI research quality along the directions listed above is not satisfactory because, according to the authors, the fact of "setting AGI as the goal" itself (regardless of how we pursue this goal) is inherently (1) conflicting with the author's position (that goals of AI research must be plural, not unified), (2) associated with hype, (3) not explicitly targetting social good in the goal.

**Position:**

Yes

**Position In Title:**

Yes

**Related Work:**

2

**Strengths And Weaknesses:**

This paper discussed about a fashionable topic, AGI, which is gaining much atttention recently across the sociaty. The many issues in current AGI works as discussed in this paper are indeed important problems from academic research perspective (and are to be addressed in practice), in my opinion. I however do not found this paper too interesting for me, for the following reasons:

1. First of all, it's a bit unclear to me what "AGI is the north-star goal of AI research" -- the position this paper is arguing against -- exactly means. After reading the whole paper, it appears to me that by this metaphor the authors wanted to refer to the opinion that "AGI should be the ultimate goal of *all* AI research". This is a radical position, and the paper should have more clearly articulated about it if this is really what the authors wanted to oppose to. Otherwise if the paper is opposing to a more modest position (such as "AGI should be the goal of *some* or *someone's* AI research"), it's even more important to clearly and accurately state the opposing position.

-- Post-rebuttal comment -- It's still unclear what "AGI is the north star" really means. Does it mean to motivate face recognition research with its potential role in AGI, instead of with the real-world demand for face recognition (e.g. as a log-in method)? Does it mean to motivate machine translation research with its potential contribution to AGI, instead of with its utility for foreigner travelers and multi-lingual websites/meetings?


2. Similarly, the paper presented little about how the "north-star position" has been adopted in the AI community. Who have advocated that in what context? The only relevant literature I can find is that in (Morris et al. 2024), an ICML'24 position paper entitled "Levels of AGI for Operationalizing Progress on the Path to AGI", which the paper under review here heavily cited, Morris et al. claimed "*achieving human-level 'intelligence' is an implicit or explicit north-star goal for many in our field*". But after a quick scan of that paper, I sensed that (Morris et al. 2024) could be just asserting the existence of an AGI community and used this fact to motivate their paper which is about *how* that AGI community should move forward. My perception is that, while now there is indeed an active group of people in AI community who indeed treat AGI as the goal of *their* works, the vast majority of AI works are still motivated by specific application scenarios.

-- Post-rebuttal comment -- The rebuttal mostly provided some evidences to show the existence of an AGI community (and their impact), which is not in disagreement. What's to be clarified in this concern is how the entire AI community is adopting the so-called "north-star" position. For example, if treating AGI as north star means specific research domains such as face recognition or machine translation are turning to motivate their work from the AGI perspective, then an investigation of such literature would be helpful.


3. A bigger concern of mine is that, while the paper being reviewed is discussing about a goal setting problem, the bulk of the paper focus on critizing how existing efforts have been (inappropriately) pursuing the goal of AGI. While these issues may exist, perhaps in a manner that the situation for AGI is worse than other AI branches (which the paper tried to argue), but why do these facts entail that people should abandon AGI *as a goal*? How do we know that the "bad practice" here is not because a particular group of related people are not doing things well, but is fundamentally rooted from setting AGI as the goal? As the authors admitted, most of the issues here are common across the entire AI community, should we stop working on AI at all? In general, I feel not convinced that the methodological issues identified in this paper well support a position about goal setting.

-- Post-rebuttal comment -- The rebuttal acknowledged that the paper does not sufficiently support the premise that "*issues [discussed in the paper] stem from AGI as a goal*". I think this is a critical premise that needs to carefully evaluated, for the purpose of this paper (not in future work). The premise that "AGI is associated with the methodological issues", as the rebuttal stressed, are not enough to support a position about (de-)motivating AGI as a goal. Issues at methodological level usually support discussions about why an existing effort to pursue a goal has been failed, and/or about how to make changes to better pursue the goal.


4. In the last section of the paper, the authors gave a short response in page 8 to the alternative view of "let's maintain AGI as the goal but improve the practice". I found this response much more relevant to the main thesis of this paper, yet the arguments here are not very inspiring to me either, possibly partially due to its brevity. For example,

- the paper argues that (even with improved practice) the goal of AGI itself is unfavorable because it conflicts with the author's opinion that AI research goals should be specific and plural. This sounds like a tautology to me -- should the community reject the AGI position with the author's opinion accepted preemptively? Why does AI research goal have to be targetting specific scenario and in plural form? The paper mostly justifies specificity and plurality with the methodological issues, but if improved research practice on AGI can address these issues, then there is no obvious reason to favor plurality over AGI (even in this paper's narrative).

- the paper argues that AGI is associated with hype so should be abandoned as the "north-star goal". But the paper does not investigate how AGI is being hyped, which part is hype and which is real progress (unless the authors think that there is only hype no progress, which is again a radical position that itself needs to be clearly articulated). It's important to think carefully here because perhaps every influential new technology comes with a level of hype. I'm not sure that an academic community should abandon a meaningful research goal just because some commecial companies hype it in their charter or somewhere.

- the paper argues that if AI really has to have a unified goal, a better alternative (compared with AGI) is "for the benefit of human". This sounds again tautological to me because arguably any research in any academic discipline is utimately for the benefit of human. Technology research should have technical goals.

-- Post-rebuttal comment -- Again, the rebuttal did not aim to address the major concerns here. Since this part of the paper is arguing to reject AGI as a goal just because it is not plural and specific, then "AI as a discipline has to target plural and specific goals" is a critical logic premise that needs to be carefully evaluated here (not in future work). The brevity of "alternative view" section is not a valid excuse because discussion on this issue should probably not be positioned as an alternative view in the first place -- the argument that "we should stop treating AGI as the north star because it's at odds with the principle of plurism/specificity" is something directly relevant to the central position of this paper *regardless of* whether the methodological issues can be resolved (which is the alternative view) or not.


5. The paper complained about the ambiguity in AGI's definition. While I appreciate the paper's elaboration on this in appendix A, I notice that some of the diverging definitions listed in Table 1 were not prorposed for defining AGI, but were raised when people argue about what "intelligence" is. I thought the letter "G" in AGI is meant to clarify that the emphasis of AGI is really on generality. Subject to the ambiguity over what "intelligence" means, AGI aims at making a single agent that has enough competency on a general enough portfolio of intelligence-related tasks -- I feel this is more or less a converging understanding of AGI.

-- Post-rebuttal comment -- The rebuttal stressed that the investigation is not just for AGI but for "AGI and related concepts". It however feels a bit unfair to associate various "related concepts" to the term AGI while on the other hand challenging its diverging definition. With similar criterion I wonder if many concepts in AI, such as "ML", "RL", "CV", "NLP", and the term "AI" itself, may all have diverging definitions (when taking "related concepts" into account).

**Support:**

1

---

> ### Author Rebuttal · Authors · 2025-04-01
>
> We thank the reviewer for their detailed and candid critique of our work, which will strengthen our arguments. Our revised version will be updated following the given concerns:
>
> 1. To further clarify the premise of our position:
>
> - The idea of "north-star" we intend is that it helps navigate a sea of possibilities. Pursuing AGI is a navigation choice that shapes AI development.
> - We don't intend to argue against the extreme position. Rather, we are recognizing the goal of AGI as having pervasive influence on research priorities, funding decisions, and public understanding of AI (eg, L 36-37).
>
> 2. We agree the paper is stronger with further details on how AGI is a north-star goal. Specifically, we contend that AGI has had outsized effects on envisioning and imagining what AI can and should be, among technical and non-technical audiences alike, due to disproportionate attention from industry leaders, governments, and media outlets. Our position is grounded on the observation that there is a growing body of research (and commentary) treating AGI as the guiding purpose behind developing increasingly advanced AI systems. We will expand the following in an Appendix, linked in the main text to better situate our position. The 'north-star' of AGI and similar concepts is promoted by:
>
> **Influential research labs and organizations**
>
> a. OpenAI
> - Mission Statement: "...to ensure that artificial general intelligence benefits all of humanity. [...] We are building safe and beneficial AGI, but will also consider our mission fulfilled if our work aids others to achieve this outcome."
>   - "Planning for AGI and beyond", 2023
>   - “Security on the path to AGI”, 2025
>
> b. Google DeepMind
> - Vision Statement: “...artificial general intelligence…has the potential to drive one of the greatest transformations in history.”
> - Lead Demis Hassabis: “...a lot of those capabilities will start coming to the fore and we’ll start moving towards what we call artificial general intelligence.”
> - “Position: Levels of AGI for Operationalizing Progress on the Path to AGI”, Morris et al. 2024
>
> c. Anthropic
> - Lead Dario Amodei in "Machines of Loving Grace" explains "Powerful AI" aligned with "AGI" goals; statedthat AI that’s ‘better than almost all humans at almost all tasks’ can emerge shortly, CNBC.
> - Recommendations to OSTP for the U.S. AI Action Plan: “We expect powerful AI systems will emerge [with] intellectual capabilities matching or exceeding that of Nobel Prize winners…”
>
> d. Microsoft
> - "Sparks of AGI", Bubeck et al. 2023
>
> e. xAI
> - Lead Musk: “The overarching goal of xAI is to build a good AGI"
>
> **Highly-cited thinkers**
>
> Frame the concept as a long-term goal, often in ways that cast AI research as counter to human interests, eg:
> - Nick Bostrom, "Superintelligence: Paths, Dangers, Strategies", 2014
> - Yann LeCunn  explains he refers to "what people call 'AGI'" as "human-level intelligence" and states “the mission of FAIR [Meta’s Fundamental AI Research team] is human-level intelligence.” TIME.
> - Yoshua Bengio: ”[Companies] compete with each other, and… think that the first to arrive at AGI will dominate.” CNBC.
> - Geoff Hinton "is certain we will have AGI soon, and biological humans will be relegated to be the second-smartest species on the planet.” Forbes.
>
>
>
> The publicity around AGI further generates significant focus on AGI as a goal within AI research. Also see Reviewer XPzd rebuttal, Q4, for link between OpenAI and Anthropic goals.
>
> 3. We deeply appreciate this point. Our premise is that the goal of AGI and methodological issues are linked, but whether issues stem from AGI as a goal, vice-versa, or the two have developed in tandem, deserves further exploration. Our proposal of "traps" functions to highlight inherent issues where potential harms aren't addressed because AGI as a conceptual framework supports less (not more) attention to existing issues within AI research. We will note the potential relationships for future work.
>
> 4. We thank the reviewer for understanding limitations of a brief Alternative Views section.
> - We will provide references to further motivate the critical role of specificity and pluralism in research.
> - Distinguishing "hype" from "real progress" may be a somewhat different paper, but we are not proposing that pursuing AGI might not produce desirable technical advancements. The response in 2. further clarifies, and the additional Appendix should strengthen this point.
> - We agree that technology research must be guided by technical goals. However, the AGI narrative postpones the setting of scientifically grounded technical goals (see "Generality Debt").
>
> 5. Table 1 is intended to capture AGI "and related concepts" (L 1156)--We will remove the Legg & Hutter reference. If the reviewer’s definition of AGI was widely adopted then we agree there may be a way to work with it to improve AGI research, but as the reviewer notes there is lack of consistency.

---

> > ### Comment · Reviewer_XaZq · 2025-04-07
> >
> > See post-rebuttal comments in the revised review.
> >
> > [Posting on behalf of reviewer XaZq so authors can respond]

---

> > > ### Author Response · Authors · 2025-04-09
> > >
> > > 1. “It's still unclear […]”
> > >
> > > To clarify: No one who navigates by the astronomical North Star expects to reach outer space, but they expect the North Star to help orient their travels toward a destination on Earth. Thus, we argue that AGI serves as a poor “guiding star” to be orienting our AI research “travels” towards. We will incorporate this language into the revised paper for further clarity.
> > >
> > > *Facial recognition*
> > >
> > > - We are not making this claim, though we show (L248-252) how exclusion (§2.6) impacts the performance of such systems.
> > > - AGI exacerbates this and similar problems by prioritizing the achievement of AGI via technical standards and definitions over mitigating unintended real-world consequences.
> > > - Treating this issue as a methodological issue misses our point: This problem could have been avoided (or identified earlier) if impacted communities had been included in design/testing.
> > > - This is a problem with the pursuit of AGI as an orienting goal within the field since AGI-related research activities focus on technical goals over social considerations, with the assumption that achieving AGI will lead to transformative solutions to many social problems, including bias.
> > >
> > > *Machine translation*
> > >
> > > - We are not claiming advancement of machine translation is motivated by AGI instead of concrete user needs.
> > > - Yet, good performance on the FLEURS dataset, for example, does not mean a model is useful (e.g., to foreign travelers). This is what we mean by Goal Lottery (§2.4): Achieving a technical goal does not imply the goal was worthwhile. That is, benchmark SOTA-chasing incentivizes optimizing for model performance under controlled conditions, where external validity must be independently assessed after the fact.
> > > - Use of similar benchmarks in AGI reflect similar lack of specificity in goal-setting which then requires a future process, assumed to have external validity, to achieve practical goals (see §2.4, L211-220 and §3 R1).
> > >
> > > 2. “The rebuttal mostly provided […]”
> > >
> > > - We are not claiming the entire AI community is adopting AGI as a north-star. Rather, we argue this community is guided/oriented by AGI, as we note above here and in our original rebuttal.
> > > - Given disproportionate resources and attention from leaders/organizations with influence on research priorities, current pursuit of AGI as an explicit goal has a direct effect on what other researchers see as the direction of AI research (i.e., the desirability of AGI-oriented/inspired research).
> > >
> > > 3. “The rebuttal acknowledged that […]”
> > >
> > > - This restatement of our rebuttal is misleading, as this is not "the premise" we assert.  Not all methodological issues are resolvable, and, as a result, achieving certain worthwhile goals may not be possible.
> > > - Thus, we do not believe pursuit of AGI is helpful as an orienting goal since it leaves evaluation of feasibility given specific tasks/goals/problems to future researchers (see §2.4).
> > > - We agree methodological issues often inspire novel solutions, but our position is that we cannot assume a solution will be found, nor can those pursuing AGI.
> > > - Given the resources devoted to pursuing AGI, and broad and/or ill-defined claims about AGI’s potential, we find this issue to be serious and significant enough to warrant our Position.
> > >
> > > 4. “Again, the rebuttal did not aim to address […]”
> > >
> > > - Our argument is not as stated here (i.e., “we should stop [...] because it’s at odds with the principle of pluralism/specitify”) so we cannot speak to the logic referenced.
> > > - We also do not claim that rejection of AGI as a goal is based on its lack of pluralism/specificity, nor are we arguing that “AI [...] has to target plural and specific goals” as both imply an unary set of goals (i.e., AGI is X where X is pluralism/specificity), which, we note, is counter to the pluralism we recommend.
> > > - We understand the reviewer does not agree with our premise here, but we want to note our argument is self-consistent in this respect.
> > > - We appreciate the reviewer’s strong engagement with our alternative view and look forward to similar conversations, as we expect many alternative views are held about this topic.
> > >
> > > 5. “The rebuttal stressed that the investigation is not just for AGI […]”
> > >
> > > - AGI’s many definitions and related concepts stand out as imprecise given the field’s past ability to navigate similar terms of art without creating similar levels of confusion about what is meant.
> > > - Given this, it is fair to link these concepts and we again note the reviewer's definition is more precise than current definitions. Further:
> > > - AI has similarly overloaded or polysemic meanings that are value-laden (i.e., how intelligence is defined and measured), yet interrogation of this term’s appropriateness continues in parallel with major progress in AI research.
> > > - To this end, we believe similar interrogation of AGI is warranted since its definitions and related concepts are contested, and thus resolving significant disagreements is imperative (L79-83).

---

### Official Review · Reviewer_u8hJ · 2025-03-17

**Significance:** 3
**Argument Clarity:** 4
**Rating:** 4
**Confidence:** 3

**Questions:**

How do you define "pluralism" in AI research in practical terms?

**Discussion Potential:**

3

**Paper Summary:**

This paper argues that the AI research community should stop treating Artificial General Intelligence (AGI) as the north-star goal of AI development. The authors highlight six key "traps" that AGI discourse creates, including illusion of consensus, supercharging bad science, presuming value-neutrality, goal lottery, generality debt, and normalized exclusion. The paper critiques the hype surrounding AGI and calls for a shift toward real-world, evidence-based AI research that prioritizes human benefit over speculative AGI ambitions.

**Position:**

Yes

**Position In Title:**

Yes

**Related Work:**

3

**Strengths And Weaknesses:**

Strengths:
The paper presents a clear and well-structured argument against the dominance of AGI as the ultimate goal of AI research.
It identifies six key issues ("traps") in AGI discourse, providing strong reasoning and examples for each.
It is likely to inspire discussion, especially among researchers interested in AI governance, ethics, and the future of AI development.
The argument is supported with references to existing AI literature and critiques of AGI claims, ensuring the discussion is well-grounded.

Weaknesses:
I didn't see obvious weaknesses in the presentation of the author's argument.

**Support:**

3

---

> ### Author Rebuttal · Authors · 2025-04-01
>
> We thank the anonymous reviewer for appreciating our argument and reasoning as well-structured and well-grounded, and for recognizing that this piece is likely to inspire discussion among researchers interested in AI governance, ethics, and the future of AI development. We agree that our definition of pluralism in AI research could benefit from greater specificity, and will update the revised draft accordingly.
>
> Regarding the question of our definition of pluralism: Practical, action-oriented meanings of the term  arise from diverse input about appropriate goals, each with their own implications:
>
> a. Methodological pluralism (Midgley, 2000; Veit, 2020; Zhu, 2022) implies that different ways of approaching a problem help achieve better solutions, often an effective strategy for complex problem-solving in general.
>
> b. Value pluralism, as applied to alignment research, implies a direct connection between technical advancement and accommodation of plural values (Sorensen et al., 2024a;b).
>
> c. Algorithmic pluralism (Jain et al., 2024) addresses the “patterned inequality” associated with algorithmic monoculture and implies a “plurality of paths to different outcomes” must be supported to avoid reproducing existing social inequalities (e.g., embedded in data).
>
> In each case, a sociotechnical approach to advancing pluralism exists in the form of practical research activities, like choice of method(s), setting of pluralistic goals, and testing algorithms to ensure known harms (i.e., algorithmic discrimination) are addressed. To name just one  example, algorithmic decision-making in hiring processes are unlikely to benefit from AGI as much as from targeted solutions to that particular setting, which account for the domain-specific nature of most positions, the different ways hiring managers evaluate candidates, and existing evidence of algorithmic discrimination in hiring.
>
> We thank the reviewer for requesting elaboration on this point. We agree our position would be stronger if we include the above points; we will add these to the revised version of the paper. Specifically, we will highlight how the work cited above (some of which was already cited in the submitted draft) supports practical strategies.

---

### Decision · Program_Chairs · 2025-04-30

**Decision:**

Accept (poster)

**Comment:**

AGI has become a rather ill defined holy grail of a lot of AI research and the narratives surrounding it. This paper does a fair job of arguing why this is detrimental to the field and provides some alternatives. It should generate good discussion.

The reviewers are generally positive. Even the reviewer giving it a low score writes a thoughtful review which, to me seemed to be more about disagreement with the position and its scope. My sense is that this shouldn't be a criterion for rejection of the position papers.

Note: the running title is missing and the ICML footer is missing. Please ensure you are using the ICML 2025 style files.